# Strategies Adopted by Addiction Facilities during the Coronavirus Pandemic to Support Treatment for Individuals in Recovery or Struggling with a Substance Use Disorder: A Scoping Review

**DOI:** 10.3390/ijerph182212094

**Published:** 2021-11-18

**Authors:** Divane de Vargas, Caroline Figueira Pereira, Rosa Jacinto Volpato, Ana Vitória Corrêa Lima, Rogério da Silva Ferreira, Sheila Ramos de Oliveira, Thiago Faustino Aguilar

**Affiliations:** 1School of Nursing, São Paulo University, São Paulo 05403-000, Brazil; pereiracf@usp.br (C.F.P.); rosamjvolpato@usp.br (R.J.V.); anavitorialima@usp.br (A.V.C.L.); sheila.oliveira@usp.br (S.R.d.O.); thiagoaguilar@usp.br (T.F.A.); 2Nucleon of Addiction Nursing Research, School of Nursing (NEPEEA), University of São Paulo, São Paulo 05403-000, Brazil; rogerio_30ferreira@yahoo.com.br

**Keywords:** COVID-19, substance-related disorders, health facilities, telemedicine

## Abstract

This review aimed to identify and synthesize strategies and actions adopted by addiction facilities to support and maintain treatment during the coronavirus disease 2019 (COVID-19) pandemic. A scoping review was conducted using the following information sources: Virtual Health Library, SCOPUS, Web of Science, PubMed, CINAHL, and Latin American and Caribbean Health Science Literature. From a total of 971 articles, 28 studies were included. The strategies to maintain the care offer were telehealth/telemedicine, counselling/screening, 24-h telephone, webinars, conducting group therapy and support among users, adaptation for electronic health records, increased methadone/naloxone dispensing, restriction in the number of medication dispensing/day, and electronic prescription and home delivery medications. These strategies can be used to support health professionals in addressing the impact of the pandemic on the treatment of those in recovery or struggling with a substance use disorder when in-person treatment is not possible.

## 1. Introduction

In 2020, the world went through several transformations due to the spread of the new coronavirus (SARS-CoV-2), leading the World Health Organization (WHO) to declare coronavirus disease 2019 (COVID-19) a public health emergency in January 2020 [1]. A pandemic was declared in March 2020 [2] and followed the same patterns as previous pandemics [3]. In addition to physical symptoms, COVID-19 also caused several psychological outcomes, such as depression, anxiety, suicide, self-harm, and increased consumption of alcohol and other drugs [4,5]. Furthermore, socioeconomic and political aspects were aggravated by structural inequalities and inequities [6], leading to increased poverty, misery, unemployment, and hunger. These factors are more serious and complex for people who struggle with substance use disorders (SUD), not only because they increase the risk of COVID-19, but also because they cause an increase in vulnerability to the disease [7]. Moreover, the sharing of substance use equipment among users, risk behavior, and constant contact with other people in scenes of use or in shelters mean that they face a greater risk of contracting COVID-19 [7,8].

Thus, when measures such as social distancing were implemented to mitigate the COVID-19 pandemic, these strategies limited the access and continuity of monitoring in the intra- and inter-sectoral networks. Therefore, during the pandemic period, several services were reduced or stopped functioning, the extent of which varied according to the program area. As stated by the Pan American Health Organization (2021), mental health services in the Americas were reduced by 77% compared to other healthcare services and programs [9].

These phenomena may also affect the care of people who struggle with SUD as access to healthcare systems was already one of the main barriers for this population even before the global pandemic [10]. Furthermore, stigma and marginalization also increase these difficulties, negatively impacting programs and policies, such as those around needle sharing, medication-assisted treatment, and other programs for the SUD management [11]. In this sense, contexts, such as primary healthcare and specialized services, are essential to addressing these phenomena as they involve prevention, diagnosis, and care for health problems, serving as a gateway to treatment for SUD patients who have a higher mortality rate when infected with COVID-19 [7,12,13].

Considering that the pandemic brought significant changes to daily life worldwide, due to social isolation and the risk of infection from COVID-19, SUD facilities might have experienced difficulties in maintaining the usual patient treatments, particularly as routines were changed, and programs and services had to reinvent themselves or change their function to readjust to the COVID-19 context. Currently, there is a need to identify which interventions and resources the SUD facilities used to maintain care provision. People with SUD already present a range of vulnerabilities, including inadequate access to healthcare. The identification and elaboration of an overview allows a wide view of the available resources in different care contexts offered during the pandemic. This review aimed to answer the following question: Which strategies (actions) were adopted by SUD treatment facilities to maintain treatment and deliver care for patients struggling or recovering from SUD during the COVID-19 pandemic?

As such, this review aimed to identify and synthesize the strategies and actions adopted in SUD treatment facilities to maintain treatment and deliver care for patients with SUD during the COVID-19 pandemic. This review intends to serve as a basis for health professionals and services that provide care for people with SUD, presenting different perspectives and possibilities of care in various contexts and countries. This will guide the use of strategies that can improve the provision of care for individuals with SUD.

## 2. Materials and Methods

This is a scoping review, with no registered protocol. It followed the standards and items proposed for scoping reviews (PRISMA-ScR) [14,15]. The strategy used was PCC (acronym for P: Population = people with substance-related disorders; C: Concept = strategies and/or actions adopted by SUD treatment facilities to keep patients in the treatment; C: Context = COVID-19). The study question was: What strategies (actions) were adopted by SUD treatment facilities to maintain treatment and deliver care for patients with substance-related disorders during the COVID-19 pandemic?

### 2.1. Search Strategy

The search strategy aimed to locate both published and unpublished studies. An initial limited search was undertaken in PubMed and the Virtual Health Library (VHL) to identify articles on the topic. The text words contained in the titles and abstracts of relevant articles and the index terms used to describe the articles were used to increase the sensitivity of the search strategy. The search strategy was developed with the following terms in PubMed: “COVID-19” OR “SARS-CoV-2” OR “2019 Novel Coronavirus” OR “Coronavirus Disease 2019” AND “Mental Health Services Administration” OR “Substance-Related Disorders” OR “Substance Abuse” OR “Alcohol Use Disorders” OR “Substance Use Disorders” OR “Drug Overdose.” Moreover, it was adapted for each information sources: VHL, PubMed, CINAHL, SCOPUS, Latin American and Caribbean Health Science Literature (LILACS), and Web of Science, aiming to identify studies on the subject; and grey literature in the information sources: MedNar-deep web search engine, CAPES and ProQuest theses, and a dissertations catalog with the objective of identifying guidelines, manuals, dissertations, and theses.

During the study selection, all studies identified were collated and uploaded into Mendeley^®^ software (Elsevier, London, UK) and duplicates were removed. Searches in electronic information sources were carried out from 25 February to 29 April 2021.

### 2.2. Eligibility Criteria

The inclusion criteria were articles addressing strategies (actions) adopted by health professionals in SUD treatment facilities to maintain treatment and deliver healthcare for patients struggling or recovering from SUD during the COVID-19 pandemic. Articles published in English, Spanish, or Portuguese were considered.

This scoping review included both experimental and quasi-experimental study designs, including randomized and non-randomized controlled trials, before and after studies, and interrupted time-series studies. Analytical observational studies included prospective and retrospective cohort studies, case-control studies, and analytical cross-sectional studies. Descriptive observational study designs included case series, individual case reports, and descriptive cross-sectional studies. Additionally, qualitative studies, letters to the editor or opinion essays, theses and dissertations, guidelines, and manuals were included as well.

### 2.3. Selection of Studies

The study selection was conducted by reading each individual title and abstract, and articles that presented indications of fulfillment of selection criteria were included in this phase. A second phase involved the reading of the full text of each article, with the analysis of criteria and their definitive inclusion in this study. In case of disagreement on inclusion, the authors had a discussion and reached a verdict; if disagreement persisted, another reviewer was invited to make a decision.

Data extraction was performed by elaborating a chart where we extracted the following variables: title of the study, year of publication, type of study, correspondence author’s country, the aim of the study, and the main strategies implemented. This chart served as guidance for allocating and classifying strategies based on their similarities. Initially, the content described in this study was divided into two main categories, and each category has two subcategories: (1) Strategies to support the care offer maintenance (telehealth/telemedicine support, adequacy of prescription, and distribution of medications); and (2) strategies that limited care offer (reorganization of face-to-face healthcare services, care aimed at preventing COVID-19, and harm reduction).

## 3. Results

A total of 971 studies were identified; 148 duplicates were excluded: VHL (*n* = 6), SCOPUS (*n* = 233), Web of Science (*n* = 66), PubMed (*n* = 601), and CINAHL (65). Another 524 were excluded during title and abstract reading, resulting in 299 articles being chosen for full reading, of which 28 met the study inclusion criteria (Figure 1).

All articles were published in English by 2020 (61%). The majority (42%) were opinion articles, followed by cross-sectional studies and experience reports (14%), case studies (11%), and the remainder (23%) adopted different methodologies. Most (68%) of the corresponding authors were from the United States, followed by Canada (6%), and Spain (6%) (Table 1).

Each article could have more than one strategy. According to the results of 13 articles, one to two strategies adopted during the pandemic were found [18,19,21,23,25,27,33,35,36,37,39,42]; 12 articles presented between three to four care strategies [16,17,20,28,29,30,31,32,34,38,40,41], and three had five or more strategies [22,26,43]. For the presentation and descriptive analysis of the strategies, they were classified into two main categories, and each category had two subcategories according to the nature and characteristics of the action: (1) strategies to support the care offer maintenance (78%): telehealth/telemedicine support (68%) and adequacy of prescription and distribution of medications (46%); (2) strategies that limited care offers (68%): reorganization of face-to-face healthcare services (39%) and care aimed at preventing COVID-19 and harm reduction (43%) (Table 2).

To better characterize the population served by the services, we identified individuals according to their place of residence and divided them to the categories of strategies used for SUD treatment. It was identified that 68% (*n* = 19 in 28 articles) did not detail this characteristic represented in this review as “living not specified.” It was possible to identify three types of living: people who live in residential facilities/residential communities (*n* = 5), homeless people/room occupancy (*n* = 5), and rural population (*n* = 5). Telehealth/telemedicine support was the most presented strategy among the population assisted by the SUD services, except among the homeless people/room occupancy, where care aimed at preventing COVID-19, and harm reduction strategy was presented (Table 3).

## 4. Discussion

This review identified and synthesized the literature on strategies adopted in SUD treatment facilities during the COVID-19 pandemic to support and maintain patient treatment. Most of the studies were opinion articles. Despite the low quality of evidence, opinion articles can be a great source of guidance for professional practice when there is an absence of research studies on a particular subject [44]. In addition, the articles originated mainly in the United States, which focused on strategies adopted in opioid use disorder treatment facilities, which might be related to the opioid epidemic in the United States [31]. Seven different countries participated in this study, pointing to a diversity of healthcare systems that might have favored the variety of interventions implemented by health services. This factor can influence access to health services, due to several facets such as financial conditions, provision of care by health providers, users’ ability to perceive care, facilities, and difficulties in reaching the health service [45]. This leads to another peculiarity: the presence of cultural diversity among the selected studies. Samhsa [46] pointed out that cultural and individual singularities can influence clinical individuals’ experiences during treatment, and in the same country, there are racial and ethnic minorities that find it difficult to access quality services. Despite these diversities, the proposal of this study was an overview of the interventions implemented by the health services to maintain the continuity of care for individuals undergoing SUD treatment during the COVID-19 pandemic, with a focus on evaluating health policies and cultural or religious peculiarities. This assessment would require the entry of complex dimensions, which can be explored in future studies.

With the restrictions imposed by the COVID-19 pandemic, health services had to remodel their activities to avoid treatment interruption. Several strategies have been adopted to maintain and support healthcare delivery. As is well known, treatment discontinuance can further aggravate SUD [47]. The strategies found highlighted that not all interventions favored the treatment maintenance of people with SUD. Due to the COVID-19 restrictions, even being essential to mitigate the spread, they negatively impacted the care offer. Among the adopted strategies, the most cited in the articles were telehealth/telemedicine (*n* = 19 in 28 articles), followed by adequacy in the prescription and distribution of medications (*n* = 13), reorganization of face-to-face care (*n* = 11), and care aimed at preventing COVID-19 and harm reduction (*n* = 12).

### 4.1. Strategies to Support the Care Offer Maintenance

#### 4.1.1. Telehealth/Telemedicine Support

Mandatory physical distancing due to COVID-19 impacted the offer of specialized health services which favored the implementation and use of telehealth worldwide [34,48,49]. Other care activities used this technology, such as counseling/screening, group therapy, and support, and the use of a 24-h telephone line, the adaptation of electronic health records, webinars, and the provision of a telephone device to users were used as part of these strategies [16,17,19,20,22,23,24,27,28,30,31,32,34,35,37,38,40,41,43]. The range of this technology is wide and diverse, favoring the maintenance of care for people who live in residential facilities/residential community [16,17,38,41,43], homeless people/room occupancy [17,19,23], and rural population [31,37,38,41]. Before the COVID-19 pandemic, the telehealth tool was used by professionals for clinical meetings, but its use for care had not been widely adopted [19,34]. Uscher-Pines [49] indicated that in some specific cases, this technology was used by professionals on trips, when moving to another city, and in other specific health situations. Studies uncovered in this review [19,20,35] found that telemedicine consultations and counseling were important for accessing and maintaining treatment, especially for individuals living in remote locations, [49,50] such as rural populations [31,37,38,41]. This resource is also favorable for healthcare providers, allowing them to have a better perception of the environment in which the individual lives [35], and in collaboration with the implementation of comprehensive care for individuals with other health conditions or comorbidities [12]. The evidence suggests that the adoption of telehealth and telemedicine may help to strengthen the bond and identify patients’ support networks, thus strengthening the potential for using this strategy in SUD treatment facilities.

For these benefits to be effective, it is necessary for specialized health services to offer different types of care, so that users can choose the option with which they feel most comfortable. Three selected studies reported that telephone care was well accepted among psychiatric patients [49], people who lived in residential facilities or residential communities, homeless people/room occupancy [17,19], and those with SUD [20]. Mahoney [51] found that the video call technology was preferred among individuals who use tobacco. Despite the use of the telephone as a tool considered to be of low cost, many individuals do not have a device or Internet access [52]. To help users access telehealth treatment, some specialized services provide a telephone device to patients [30,40]. Another viable option for addressing this situation is the distribution of prepaid cards and devices with prepaid services for patients who did not have such resources [53]. Thus, individuals could use the telephone lines for some specialized services made available 24 h a day, providing psychological support, counseling [28], and initial assessment until the substitution prescription for those with an opioid use disorder [20].

As barriers to this service, professionals identified the absence of physical contact for feedback during counseling and physical examination [34], lack of equipment, limited Internet access, and limited handling skills identified by health professionals among some clients, rural population, and people who live in residential facilities/residential community [38]. These same difficulties were also reported by users [35], including a lack of familiarity with using this new resource [34]; however, it was noticed that even individuals who lived in precarious or rural housing had a smartphone [38]. Other authors also found the same difficulties among professionals and users [12,49,50], which reflects the lack of bonding and support for achieving goals proposed during treatment [51]. To minimize the users’ barriers in telehealth access, Harris [23] highlighted that the collaboration of care entities (religious institutions) and general health services can be an essential facilitator for individuals vulnerable socioeconomically, without an access to the necessary technology. In addition to service offers, it is important for the team to be sensitive when identifying these barriers, as these details could be determinants of the occurrence of relapses, moving users away from starting treatment, or maintaining treatment during the absence period.

Support among users began to be provided by videoconference and presented positive results, with an increase in the participation of individuals in the groups [34]. This type of support is an important component for individuals in addiction treatment, acting as support, increasing self-efficacy, decreasing relapses, and helping to improve involvement with treatment [54,55]. Telehealth care was widely adopted by psychologists during the COVID-19 pandemic, but these professionals highlighted the need for specialized training for effective care, especially crisis management [56]. However, group therapies realized by telehealth can cause embarrassment among individuals undergoing treatment [35].

The webinars were used to offer training and orientation regarding COVID-19 to care providers of residential facilities for people undergoing SUD treatment living in residential facilities/residential communities [16]. Webinars are a form of continuing education presented as web-based seminars, which have constituted a major way of education and learning during the pandemic [57,58]. Some specialized treatment services offered lectures on COVID-19 and substance use as webinars to provide adequate treatment for opioid use disorders [59,60].

With telehealth care rising in many formats, electronic health records (electronic medical records) have increased on a large scale within services. Therefore, professionals had to adapt to fulfill these records with adequate and necessary information for continuity of care in a client-centered manner and to improve care [61]. This fact also caused discomfort and difficulty, as well as burnout symptoms, among nurses and physicians during the pandemic [62].

Telehealth/telemedicine support was the most used strategy to maintain care, reaching its purpose for the maintenance of the care offered by different types of professionals and diverse populations on SUD treatment. Although few studies have addressed the acceptability of this intervention, it is noticeable that, in some cases, the adherence and outcomes from the use were positive. Overall, telehealth/telemedicine support is an important resource that can become a permanent part of care in SUD treatment facilities.

#### 4.1.2. Adequacy of Prescription and Distribution of Medications

One of the strategies adopted was the adequacy of prescription and distribution of medications during the COVID-19 pandemic, such as electronic prescriptions, increased methadone/naloxone dispensing, restricting the amount of medication dispensed/day, support of primary care in dispensing medicine, increasing the supply of training for medicine distribution, decentralization of medicine dispensing, and home delivery [17,20,25,26,27,28,29,31,33,36,38,39]. This change occurred due to adjustments in guidelines in some countries, such as the United Kingdom and the United States. In the United Kingdom, until July 2021, a person could take a 14-day supply for self-administration or appoint someone to collect the medication dispensed on his or her behalf [63], and in the United States, the Substance Abuse and Mental Health Services Administration (SAMHSA) made more flexible regulations about the supply of opioids and specialized health services were encouraged to keep patients on self-administered doses (methadone/naloxone), thus facilitating the implementation of physical distancing [13,45,64]. That provided that individuals in different places of residence continued the use of medication essential to maintain the treatment in different types of care to people living in residential facilities/residential community [17,38], homeless people/room occupancy [17,25,39], and rural populations [31,36,38]. Prescriptions via telemedicine or teleconference were also allowed to have increased drug distribution limits, without the need for a face-to-face visit during the contingency plan [20,27], facilitating the maintenance of treatment for communities with difficult access, such as rural populations [31,38]. The possibility of assistance through telehealth platforms, telephone, and other sources allowed to mitigate the impact of COVID-19 on public health, corroborating the results found in the present review [64].

The dispensing of methadone/naloxone was facilitated, increasing both the number of doses available and the number of individuals benefiting, as well as reducing harm in relation to overdoses [28,29,33], to individuals who live in rural communities and in residential facilities/residential communities [36,39]. Previously, the unsupervised use of methadone was allowed, as long as these individuals showed stability in their condition, such as the absence of recent drug abuse, stability in social relationships, absence of serious behavioral problems in the clinic, or involvement in criminality, such as drug trafficking, and capacity to safely store medicines for opioid use [65]. These changes can cause concerns, such as the lack of professionals available to administer medications to reverse opioid overdose conditions in a timely manner [66] and over the sharing or selling of dispensed medications. The studies analyzed in this review [28] did not report cases of overdose or death; in rural communities, this positive effect was attributed to the ease and proximity of medication in more remote areas [36], and only 6% (*n* = 87) of cases of medication sharing were observed [33]. Strengthening precautions, health institutions increased the offer of training for the distribution of drugs that prevent overdose [36], and training was offered online; thus, reference individuals in the community and/or family members were also able to use the Take-Home Naloxone kit in overdose cases [67]. These data open the possibility for fostering discussion on the development of new policies and recommendations from regulatory bodies that meet the real needs of individuals, such as harm reduction and encouragement of accountability with the proposed treatment.

To reduce agglomeration in drug distribution sites, a restriction on the number of medication dispensing/day was observed [17]. This change can represent a barrier to access to medication, affecting the continuity of treatment and putting individuals at risk [53]. An alternative could be the dispensing of electronic medicine boxes, facilitating the delivery of larger quantities of medicine in a supervised manner and allowing the health team to identify the need for guidance regarding use [68].

Among the actions/interventions adopted was medication delivery at home [28], which offers the convenience of eliminating the expense of traveling and reducing stigma in relation to the demand for specialized services for medication withdrawal [19]; the act of taking the treatment to the user helps to maintain objectivity, increases acceptance and adherence to treatment [69], and reduces the risks of coronavirus contamination. Decentralization in drug dispensing and primary care support [17,25,26] even favored the most vulnerable individuals, such as homeless people/room occupancy [17], strengthening strategies to keep individuals in treatment and the bond with the community. Primary health care is the gateway to the treatment of chronic diseases, and with prescription assembly solutions (PAS) with proper support, it can effectively treat these individuals [70]. However, Lagisetty [71] pointed out that it is still necessary to formulate health policies so that care is coordinated between specialized and primary healthcare services.

The strategies were adopted to facilitate access to medication and treatment for disorders related to the use of psychoactive substances, especially for keeping these individuals protected against COVID-19. Health institutions increased the offer of training for the distribution of medications that prevent overdose, and this training for new dispensers or recycling of working professionals is carried out online. What made training and viability possible in rural areas? It was feasible to observe that the increase in the amount of dispensed drugs to Take-Home did not cause serious complications. This fact brings new perspectives on how to establish treatment with longer follow-up in certain cases that may lead to new policies and recommendations for prescribing and dispensing drugs for opioid substitution therapy (OST). These individuals may benefit from the maintenance of treatment as a decrease in stigma. However, studies with high-quality evidence to evaluate the efficiency of this new method of prescribing and dispensing OST medications are necessary.

### 4.2. Strategies That Limited Care Offer

#### 4.2.1. Reorganization in the Face-to-Face Assistance of Specialized Services

The strategies were implemented with the purpose of mitigating the spread of COVID-19 and its consequences, maintaining assistance to users with greater vulnerability, and avoiding leaving them destitute. As seen before, health providers quickly adapted their services to online and telephone care [12,72], in addition to using other remote monitoring devices [73]. For face-to-face care, studies report the need for rapid adjustment in hours of care and visits, rules of conduct, reduction in the number of patients seen, and reduction in the supply of specialized services [16,18,22,26,28,29,30,31,41,42,43]. It was observed that this intervention category was the only one that did not present specific actions to homeless people or room occupancy. These determinations imposed by regulatory agencies may have pushed a considerable number of individuals from the SUD treatment routine [74]. Care interruption may represent setbacks in care planning by health professionals with the patient, as well as losing positive connections among the services, professionals, peers, and society, which are compromised due to substance use [75].

Regarding family visits to specialized service clinics, people living in residential facilities/residential communities [16,43] with social restrictions, general health guidance was published for health providers with recommendations to avoid meeting more than 10 people, trips for social visits, and for preferably staying inside the home, especially when one family member tested positive for COVID-19 [76,77]. Some residential treatment programs in residential facilities/residential communities restricted the personal visits of family members to only when the pandemic was in the worst phase, maintaining social distance, and the use of PPE during the meeting, which resulted in some dropouts from the program [43]. Some recovery programs chose to adopt certain rules of conduct to improve the experience in the place, such as substance use [16]; Jason et al. showed that other rules can help to improve the living in recovery housing, as fulfilling the division of responsibilities among residents, motivating them to remain sober and, thus, avoiding the use of alcohol and other illicit drugs during the treatment period at home [78]. For those who received visits without authorization or violated institutional rules, quarantine and COVID-19 tests were performed for the safety of others [79].

Regarding visits by users to in-person treatment programs with opioids (OTPs), their frequency reduced with the availability of taking a certain dose of methadone home and ensuring treatment with medications for opioid use disorder [29]. Some regulations have been relaxed in the federal guidelines governing specialized health services, especially during the pandemic period, which were adopted to ensure that as few people as possible were kept within specialized health services and to ensure physical distancing [80,81].

Another change in specialized service providers was regarding adjustments in service hours [18,29,30,42], even among rural populations [32,41], which needed to adjust the period of operation of specialized services, and suffered a reduction in professional teams or adjustments in shifts and working hours [82,83]. Thus, it was possible to serve the demand of this vulnerable population and reduce the associated damages. There were also changes in the schedule of OTP visits due to travel by public transport at times that allowed greater physical distancing [29].

However, during the period of operation of specialized health services, the government’s decision to restrict the hours of operation of community pharmacies may compromise the treatment of some users in the withdrawal of medication and, consequently, increase the risk of overdose due to the use of opioids, while others also described more hours of work with alternate working days [84]. Therefore, some supply distribution locations opted to maintain longer weekend opening hours, in addition to supporting access in regions with difficult access [67]. Finally, some of the harm reduction service providers had their capacity reduced, due to the illness of workers, in many cases due to poor working conditions and increased vulnerability to COVID-19, leading to more restricted hours of operation, due to the absence of currently available professionals [42,85].

The decrease in the number of health workers also affected the reduction in the number of patients seen [22,26,42], residential facilities/residential community services presented extensive financial losses and, as a consequence, dismissal of employees, which provoked an overload among those that remained [43]. The reduction was also due to social isolation restrictions, in which patients were restricted from seeing family members for a period, which led to the abandonment of some residential programs because of their restricted personal interactions [43,86]. With the closing of clinics and substance treatment use shelters, there was a lower provision of specialized services for these people and, thus, a lower number of users being assisted. All of this added to the precarious availability and access to effective treatments, prevention strategies, and the sociocultural context [24,87].

In residential program facilities, the problem is even greater, as the new public health guidelines not only limit the number of beds that can be filled, but also require that new patients be tested for COVID-19 before entering the residence [88]. Such strategies can become a barrier in the treatment process of this population because of the lack of access to COVID-19 testing and the lack of sufficient programs that support all individuals in spaces with safe distancing, which may increase the risk of exposure [43].

The control measures adopted in specialized services resulted in a reduction in the supply of care [28,31], with fewer employee working hours, lower service capacity, reduced opening hours, fewer visits, fewer beds, reduced group treatments, and other services suspended to converge with measures to protect third parties present at the site and self-protection, while the COVID-19 pandemic persists, which, although necessary, can interfere with patient involvement in treatment [28,82,84,86,88].

In this sense, it is noticeable that the barriers imposed by the COVID-19 pandemic and the reduction in the availability of specialized services can produce long-term effects for individuals who already face poverty and physical and mental health problems [12,47], as they can provide a feeling of loneliness and can lead individuals to interrupt a treatment [38] that, for years, was the main form of therapy offered [52], in addition to the financial barriers that made several specialized services interrupt provision of care [38]. At this time, health professionals must be proactive and agile in assessing individual needs to establish the best care and management of SUD, even with the care offer reduction or care at a distance.

#### 4.2.2. Care Aimed at Preventing Coronavirus and Harm Reduction

Several strategies to mitigate the virus have been adopted in the routine care of individuals who use psychoactive substances, such as quarantine, use of PPE, hand hygiene, social distancing, and screening/testing for COVID-19, assistance to homeless people, access to material for harm reduction, approaching other health networks, and drug testing suspension [16,17,21,22,25,26,30,32,38,40,41,43]. Providing effective care, regardless of the place of living, among the main measures adopted by programs, hand hygiene was prominent [26], while other measures included individual use of articles for substance consumption [25], spaces reserved for individual use, distance between people [89], and individual actions, whether through the use of alcohol gel for hand cleaning, respiratory labeling, or surface cleaning [90]. To ensure social distancing, with the objective of reducing community transmission of COVID-19 [89], one of the services that maintained care for the homeless people/room occupancy population [25] adopted a “phone booth” model applied in an isolated site to exchange syringes and hygiene care to each user, with professionals evaluating and offering medicines to treat dependence.

Even before the pandemic, people with SUD have been identified as a vulnerable population [91,92]. Despite these efforts, this care can be deficient and increase the risk of coronavirus infection in those who are homeless or live in precarious conditions, in crowded environments, and among those who share syringes and other materials to use drugs [13,93]. These factors are aggravated by pre-existing vulnerabilities, which deserve special attention, not only because they suffered from social marginalization and stigma, but also because they lacked even more access to specialized health services, contributing to the emergence of other diseases. To minimize these problems, both in relation to the substance and COVID-19 contamination, the Castile and León Treatment of Dependence Network in Spain implemented a program that offered care to these homeless individuals confined in a shelter due to coronavirus circulation restrictions [17]. For this reason, treatment programs must adopt a system of specialized services that include strong policies with a wide range of well-structured resources, such as facilities and trained personnel, and effective, accessible, cheap, and integrated services; thus, they have been prepared not only for the current pandemic scenario, but for others that may arise [94]. The strengthening of health services must be encouraged. Harris [40] highlighted that the network care between primary care and hospitals was essential to provide therapeutic care in an integrated and shared way by both hospitals and primary services.

As indicated before, the pandemic mostly affects the most vulnerable groups [95], including drug addicts; thus, among the strategies aimed at this population should be testing for COVID-19 and quarantine for a minimum of 14 days for users who wish to enter residency services. To ensure the safety of everyone involved at the site, users, or workers, the importance of maintaining or implementing activities to treat addiction should be considered, as well as tackling the dissemination of the coronavirus with testing, which is one of the most efficient methods of control in combating the pandemic [79]. These interventions aimed to protect against the coronavirus may be an aggravating factor in the maintenance of SUD treatment. Many users and services had significant impacts on financial resources [16,19,24,34] such as, for instance, the 14 days of quarantine to patients and health professionals, which interrupted the treatment [22]. Service providers for the rural population indicated that stress due to the fear of contamination by COVID-19, isolation, and financial impact can exacerbate the use and relapse of individuals undergoing treatment [38]. These effects should be observed over time by team SUD treatment services and should be explored in future studies that can detail through surveys and appropriate instruments, the individuals who use the services.

It is noteworthy that even with the strategies adopted and detailed in this study, these individuals face years with a range of diseases related to SUD. Individuals who use psychoactive substances such as alcohol, tobacco, and cocaine presented alterations in immunologic, pulmonary, and cardiac systems [96,97,98] and have a higher chance of requiring mechanical ventilation when they develop pneumonia [99].

Reflecting mainly on lost lives or disabilities, it is estimated that in 2017, about 42 million years of healthy life were lost due to SUD [100]. The occurrence of COVID-19 and SUD in the United States and their consequences were detailed by Wang [101], who evaluated the risk of people with SUD (at life *n* = 1.880) to contract COVID-19 when compared to people without SUD diagnosis; people who do opioids were at a higher risk (2.42%), followed by cocaine (1.57%), alcohol (1.42%), and tobacco users (1.33%). The mortality rate among these individuals was 9.57%. These data demonstrate the seriousness of the total or partial interruption of specialized services.

Health professionals play an important role in providing guidance on the risks of contamination by coronavirus in identifying and directing appropriate and specialized treatment for individuals who are recovering or struggling with SUD. The aim is to reduce risks, barriers, and stigma when seeking assistance for addiction [12], as these individuals struggle with substance use and generally present mental health-related problems such as anxiety, depression, and suicidal ideation [38]. The suicide risk among individuals with SUD diagnoses is 7.13 times higher than that in the general population. Suicide can occur when the individual is under the influence of the substance or immediately after the influence is gone [102]. The mental health of these individuals is also compromised, mainly because of fragile family and support group ties [12]. This type of care is over the long term, as addiction is a chronic disease, and its identification occurs when the condition has existed for a few years [103]. In the future, we must prepare for an increase in the number of new individuals or the recurrence of problems with SUD in the post-pandemic period [38].

This review provides an overview of the actions taken by SUD treatment facilities during the pandemic period, identifying interventions with the potential to be incorporated in the care of these individuals. Some gaps were identified in the selected studies, which could guide future research. Many of the studies did not show the acceptance and adherence of individuals in relation to the implemented actions, in addition to addressing the actions without covering the exclusive difficulties of specific populations (elderly, homeless or sheltered, population in deprivation of liberty, and others). In this case, the homeless and sheltered people were already having difficulties accessing health services due to stigma or because they did not fit into social norms (having a fixed residence and/or documentation), and these items need to be cared for in health services [104]. Another point is the few studies addressing alcohol, a worldwide disorder that affects 237 million men and 46 million women, causing 7.2% of all deaths considered premature among individuals aged up to 69 years [105]. Furthermore, it was not possible to identify the number of individuals who abandoned treatment during the pandemic, which would lead to new perceptions about the actions/interventions adopted by the services. This is a relevant factor for SUD specialists to establish focused strategies for the rapprochement and strengthening of the link with the services.

### 4.3. Limitations

Finally, this study has limitations that should be addressed by future studies: methodological weaknesses in the primary sources were found in the literature, indicating that future studies could be more robust in the methodological scope; there were no data that could allow the comparison of the proposed and adopted strategies and their treatment outcomes; we did not develop a review protocol previously. We may have lost the studies because of our language inclusion criteria. Another limitation is that the study focused on strategies that were implemented, and some data about recommended strategies may have been lost during the selection of the articles. Due to the methodological differences among the studies found, it was difficult to establish which strategies were better applicable to a specific population, such as homeless people with SUD. For this reason, it is necessary to conduct more rigorous studies.

## 5. Conclusions

There is little available literature on interventions adopted by SUD treatment facilities to maintain treatment and to deliver care to patients struggling with or in recovery from SUD during the COVID-19 pandemic. Most studies have focused on the care of individuals with opioid-use disorders. Despite the evidence synthesizing resulting in predominately opinion articles, they can help health practitioners and stakeholders to address the impact of the pandemic social living restrictions for those who are in treatment for SUD when in-person activities are not possible. Most strategies described in the articles were related to biological aspects of care, such as prescription and maintenance of the distribution of medication. Evidence about the successful experiences and effectiveness of psychosocial interventions involving populations undergoing treatment for other psychoactive substances, such as alcohol, during the pandemic period should be encouraged.

## Figures and Tables

**Figure 1 ijerph-18-12094-f001:**
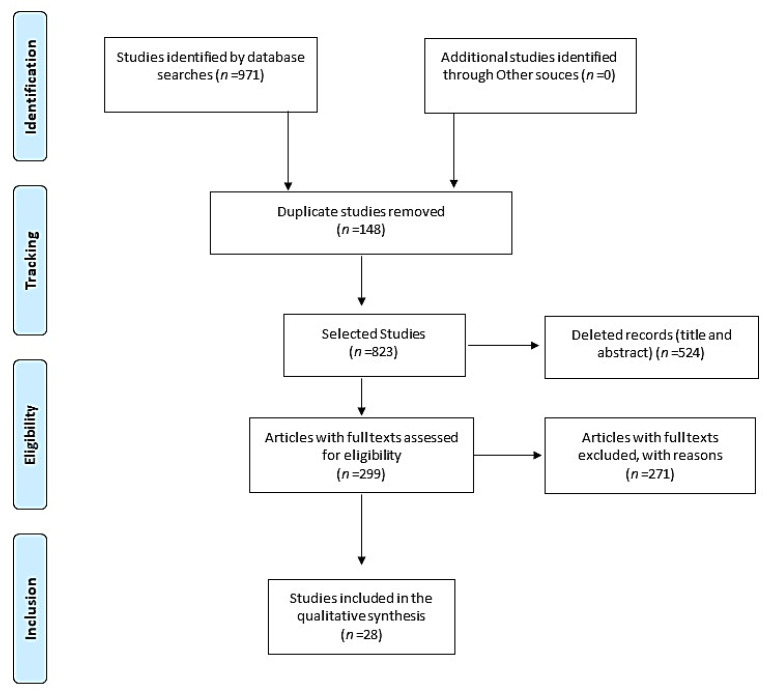
Preferred Reporting Items for Systematic Reviews and Meta-Analyses flowchart of the study identification, screening, and selection process.

**Table 1 ijerph-18-12094-t001:** Descriptive analysis of articles that propose strategies (*n* = 28).

	Number of Articles (%)
Year of Publication	
2020 [16,17,18,19,20,21,22,23,24,25,26,27,28,29,30,31,32]	17 (61)
2021 [33,34,35,36,37,38,39,40,41,42,43]	11 (39)
Type of Study/Article	
Opinion articles [16,20,21,22,25,26,30,31,32,35,36,38]	12 (42)
Cross-sectional study [17,19,28,34]	4 (14)
Experience report [27,29,40,41]	4 (14)
Case study [23,24,39]	3 (11)
Qualitative study [19,43]	2 (7)
Analytical study before and after [33]	1 (4)
Retrospective study and post hoc analysis [37]	1 (4)
Mixed study (quanti/quali) [42]	1 (4)
Country of Corresponding Author	
U.S.A. [16,19,20,21,23,24,25,27,29,31,32,33,34,35,36,38,40,41,42,43]	19 (68)
Canada [32,39]	2 (6)
Spain [17,18]	2 (6)
Ireland [30]	1 (4)
India [37]	1 (4)
Italy [26]	1 (4)
Bosnia and Herzegovina [28]	1 (4)
Israel [22]	1 (4)

**Table 2 ijerph-18-12094-t002:** Sample distribution according to category, number of studies, and strategies in the services. Brazil, 2021 (*n* = 28).

Categories	Strategies Adopted in Specialized Health Services
Strategies to Support the Care Offer Maintenance	
Telehealth/telemedicine support (*n* = 19) [16,17,19,20,22,23,24,27,28,30,31,32,34,35,37,38,40,41,43]	Telehealth/telemedicine [17,19,20,22,23,24,27,30,31,32,34,35,37,38,40,43] Counseling/screening [34,35,41] Conducting group therapy and support among users [34,35] 24-h telephone [20,28] Adaptation for electronic health records [40] Webinars [16] Support with the provision of telephone devices to users [30,40]
Adequacy of prescription and distribution of medications (*n* = 13) [17,20,25,26,27,28,29,31,33,36,38,39]	Electronic prescription [20,27,31,38] Increased methadone/naloxone dispensing [28,29,33,36,39] Restriction of the number of medication dispensing/day [17] Home delivery of medications [28] Decentralization of distribution and support of primary care in drug dispensing [17,25,26] Increased offer of training for drug distribution [36]
Strategies that Limited Care Offer	
Reorganization of face-to-face healthcare services (*n* = 11) [16,18,22,26,28,29,30,31,41,42,43]	Visits by family members to service clinics [16,29,43] Rules of conduct [16] Adequacy of service hours [18,29,30,31,41,42,43] Reduction in the number of patients seen [22,26,42,43] Reduction in the services offer [28,31]
Care aimed at preventing COVID-19 and harm reduction (*n* = 12) [16,17,21,23,25,27,31,33,39,40,42,43]	Quarantine [16,22,26,41,43] Use of personal protective equipment (PPE) [22,26,38,43] Hand hygiene/social distancing [22,26,32] Assistance to homeless people [17] Access to material for harm reduction [21,25] Approaching other health networks [40] Suspension of drug testing [43] Screening/Testing for COVID-19 [30,38,41,43]

**Table 3 ijerph-18-12094-t003:** Sample distribution according to category, number of studies, and living characteristics of the assisted population by the services and intervention categories. Brazil, 2021 (*n* = 28).

Living Characteristic	Categories
Living not specified (*n* = 19) [17,18,19,20,21,22,24,26,27,28,29,30,32,33,34,35,37,40,42]	Telehealth/telemedicine support (*n* = 13) [17,19,20,22,24,27,28,30,32,34,35,37,40] Adequacy of prescription and distribution of medications (*n* = 7) [17,20,26,27,28,29,33] Reorganization of face-to-face healthcare services (*n* = 7) [18,22,26,28,29,30,42] Care aimed at preventing COVID-19 and harm reduction (*n* = 6) [17,21,27,33,40,42]
People who live in residential facilities/residential community (*n* = 5) [16,17,38,41,43]	Telehealth/telemedicine support (*n* = 5) [16,17,38,41,43] Adequacy of prescription and distribution of medications (*n* = 2) [17,38] Reorganization of face-to-face healthcare services (*n* = 3) [16,41,43] Care aimed at preventing COVID-19 and harm reduction (*n* = 3) [16,17,43]
Homeless people/room occupancy (*n* = 5) [17,19,23,25,39]	Telehealth/telemedicine support (*n* = 3) [17,19,23] Adequacy of prescription and distribution of medications (*n* = 3) [17,25,39] Reorganization of face-to-face healthcare services (*n* = 0) Care aimed at preventing COVID-19 and harm reduction (*n* = 4) [17,23,25,39]
Rural population (*n* = 5) [31,36,37,38,41]	Telehealth/telemedicine support (*n* = 4) [31,37,38,41] Adequacy of prescription and distribution of medications (*n* = 3) [31,36,38] Reorganization of face-to-face healthcare services (*n* = 2) [31,41] Care aimed at preventing COVID-19 and harm reduction (*n* = 1) [31]

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
