# Peer review of "Strategies Adopted by Addiction Facilities during the Coronavirus Pandemic to Support Treatment for Individuals in Recovery or Struggling with a Substance Use Disorder: A Scoping Review"

_ijerph, 2021, doi:10.3390/ijerph182212094_

Round 1

Reviewer 1 Report

Thanks to editor to offer me the possibility to review this well-documented article concerning an important field of consequences in health care facilities during covid-19 pandemy. thank you to the authors for this rigorous work

Unfortunately, studies with a high-level of evidence are lacking partially limitating the contribution of this review. 

Even if the methodolgy appaers to be well organised and the synthesis well done, rational justifications could be imporved to better specified the constuction of purpose and the choice of question.

I have no special comment about the method even if, face to few litterature available, the type of article might have been discussed and the choice of systematic review questionned.

results are well presented

discussion is ok even if some points could improved it:

1/ what about cultural differencies (in regard of public health policies and health care system, we note no study from asia and very few from europe)?

2/ telehealth acceptability for persons suffering of SUD

3/ psychiatric comorbidities or psychological impact of covid 19 in people with SUD

4/ references to expert reccomandations and guidances (ex UK Nations...)

limitations could be more developped insisting on bias of selection and inclusion 

Author Response

Dear reviewer,

Thank you for taking the time to read our manuscript, we are very pleased that you enjoyed our study. We are very welcomed to read your suggestions, and we made some changes according to your comments.

Reviewer 2 Report

GENERAL COMMENTS

              This article, submitted to the “Systematic Review” section, identifies and synthesizes strategies and actions taken by substance use disorder (SUD) treatment facilities during the COVID-19 pandemic.  It is undoubtedly an issue of great interest in which the available information is scarce and its practical application is difficult, given the multiple organizations and disparity among care structures. Given that it concerns a highly vulnerable population, knowledge for adopting solutions at different levels of care is very necessary. The problem lies in always offering proven solutions for well-identified populations in a defined context. One of the limitations of this article is that it offers a catalog of strategies for very distinct patients in very different specialized services and in very diverse settings (for example, individuals who live at home and those who live in residential facilities). As we understand it, this greatly limits its interpretability and applicability.

SPECIFIC COMMENTS

  1. Though the article is identified as a systematic review, its structure is more like a narrative. It would be advisable for the authors to specify which PRISMA 2020 Checklist criteria the article meets and which it does not.
  2. The description of the time frame for the literature search is noteworthy. It is not clear what the period analyzed was or the dates of the most recent consultations. In addition, backward citation search strategies for other references are not described. Having more information on these aspects would be advisable.
  3. The context indicated in the search strategy leads to the inclusion of patients in very different situations (for example, homeless individuals or residential facility residents). This makes it so that the catalog of strategies combines very diverse actions for very different patients in entirely disparate situations. From a practical point of view, wouldn’t it be reasonable to separate them into subgroups?
  4. The inclusion and exclusion criteria are not specified in detail.
  5. There is no description of the data collection process methods. It would be of interest to include it.
  6. The synthesis methods that allowed for identifying and classifying the strategies are not described. It would be of interest to have more information on this aspect.
  7. In general, the Materials and Methods section offers little detail, which hinders the reader from knowing precisely how the study was conducted. For example, the creation of the strategy categories is a bit confusing, as in its current formit mixes strategies to support treatment with others that reduce the service offer or which are very general. Although the latter two may have been imposed due to the necessities of managing the COVID-19 pandemic, mixing them does not seem reasonable given the study’s aim.
  8. Likewise, the distribution of strategies into categories is a bit confusing. For example, it is not clear why educational webinars are not included in the "Telehealth” section.
  9. Among the strategies analyzed in the “Telemedicine” section, very notable differences are detected in their design, patients included, objectives sought, etc. For example, studies with outcomes (for example, reference 27) are included with others that merely describe the actions (for example, reference 40). Likewise, studies centered on patients with SUD (for example, the aforementioned references 27 and 40) are included with others that describe very general experiences and in which patients with SUD represent a small part of the sample (for example, reference 28). Therefore, would the results be different with more homogeneous sample?
  10. In Table 2, reference 28 is not included in the Categories column but is included in the Strategies column. Could this be an oversight?
  11. Given the issues raised, it seems reasonable to include a Limitations subsection in the Discussion section.

Author Response

(The authors gave the same response as above.)

Reviewer 3 Report

The paper is well written and clearly structured; the topic is of course relevant and timely Having said that, in reviewing this paper I would have expected to be presented not only with the different studies but also with the limitations and risks of the different strategies adopted, of course from the Authors’ point of view. An example: when debating about the increase in take away dosages of methadone and buprenorphine to allow clients to spend more time at home without the need to go to the pharmacy one should consider as well that most clients were given 14-day take away dosages. These dosages amounted to quantities exceeding 1 litre, and at times 2 litres, of methadone per single dispensing occasion. Indeed, a range of overdoses and deaths were observed as a result of this and papers should be published on these issues.

Furthermore, during the pandemic a range of stigmatisation issues have been observed; clients suffering from very severe drug or alcohol withdrawal not transported with the ambulance to the hospital because … they were not COVID cases. From the medical point of view, when discussing about addiction issues during the COVID pandemic all the above issues should be critically discussed by the authors; this would make the paper more interesting and clearly more appealing. As a reader, I would strongly value the opinion of the Authors, maybe in a specific section of the Discussion.

Author Response

(The authors gave the same response as above.)

Round 2

Reviewer 1 Report

Dear Editors and authors

thank you for your trust and for this opportunity to reviews these corrections for this scoping review.

Changes made by the authors seems me to substantially improve the article.

Even if difficulties of the purpose are that previews litterature remains limited because of the extraordinary situation that we are experiencing, an overview of the interventions implemented by the health services to maintain the continuity of care for individuals undergoing SUD treatment during the COVID-19 pandemic appears to be helpfull to guide research related to strategies and interventions in extraordinary situations as we are experiencing.

I do not have comment anymore

Thank you for this interesting article.

Author Response

Dear Reviewer

Thank you for your time and your contribution to our research. It was a pleasure. We are very grateful and pleased that you liked our article. Again, we are very welcomed for your comments and suggestions. Thank you very much.

Best Regards

Reviewer 2 Report

We have read the changes the authors made to the manuscript with great interest. Its “conversion” into a Scoping Review is wise as this allows for a more appropriate view of the findings in a field with data that is scarce, scattered, and intermingled. However, there continue to be aspects that require further explanation:

1. We do not believe that point 9 of the prior review was adequately addressed.

2. The authors should state the limitation of not having a review protocol when conducting the study in the text.

3. In a Scoping Review analysis, in addition to describing and synthesizing the available data, it is common to indicate gaps in existing knowledge. However, in this regard, the manuscript lacks a more adequate evaluation and statement of the areas in which data or insufficient or confusing.

Author Response

Dear reviewer,

Thank you for taking the time to read our manuscript.  We made some changes according to your suggestions. 

  1. We do not believe that point 9 of the prior review was adequately addressed.

To detail more precise the type of population that was assisted by the SUD treatment facilities, a table was added (Table 3.) on page 5. On this table we relate the living characteristics detailed on the articles to the intervention categories. It also added more details along the manuscript,  from page 7 to 12. We would like to point out again that the aim of our review is to identify and synthesize strategies and actions adopted by  addiction facilities to support and maintain treatment during the coronavirus disease 2019 (COVID-19) pandemic. However, your suggestions were very important to improve the characteristics aspect of the population assisted and the range of the interventions. 

  1. The authors should state the limitation of not having a review protocol when conducting the study in the text.

It was added to the manuscript: page 13; 4.3. Limitations. 

  1. In a Scoping Review analysis, in addition to describing and synthesizing the available data, it is common to indicate gaps in existing knowledge. However, in this regard, the manuscript lacks a more adequate evaluation and statement of the areas in which data or insufficient or confusing.

 It was added to the manuscript: pages 11 and 12: "This review provided an overview of the actions taken by SUD care services during the pandemic period, identifying interventions with potential to be incorporated in the care of these individuals. Some gaps were identified in the selected studies that could guide future research. Many of the studies did not show the acceptance and adherence of individuals in relation to the implemented actions, in addition to addressing the actions without covering the exclusive difficulties of specific populations (elderly, homeless or sheltered, population in deprivation of liberty, and others). In this case, the homeless and sheltered people were already having difficulties in accessing health services, due to stig-ma or because they do not fit into social norms (having a fixed residence and/or documen-tation), and these items are necessary to be cared for in health services [104]. Another point is that few studies addresses alcohol use, a worldwide disorder that affects 237 million men and 46 million women, causing 7.2% of all deaths considered premature among in-dividuals aged up to 69 years [105]. Furthermore, it was not possible to identify in the studies the number of individuals who abandoned treatment during the period of the COVID-19 pandemic, which would lead to new perceptions about the ac-tions/interventions adopted by the services. This is a relevant factor for SUD specialists to establish focused strategies for the rapprochement and strengthening of the link with the services."

Again, we are very welcomed for your comments and suggestions and, we hope we had attended your suggestions. Thank you very much!

Best Regards